# Polycyclic aromatic hydrocarbons and their metabolites in bronchoalveolar lavage and urine samples from patients with inhalation injury throughout their hospitalization: A prospective pilot study

Katerina Vyklicka[1], Petr Gregor[1], Bretislav Lipovy[2,3,4], Filip Raska[3], Petr Kukucka[1], Jiri Kohoutek[1], Petra Pribylova[1], Pavel Čupr[1], Petra Borilova Linhartova[1]*

1 RECETOX, Faculty of Science, Masaryk University, Brno, Czech Republic, 2 Department of Burns Medicine, Third Faculty of Medicine, Charles University and University Hospital Kralovske Vinohrady, Prague, Czech Republic, 3 Department of Burns and Plastic Surgery, Faculty of Medicine, Masaryk University, Kamenice, Brno, Czech Republic, 4 Department of Burns and Plastic Surgery, University Hospital Brno, Brno, Czech Republic

☯ These authors contributed equally to this work.

* petra.linhartova@recetox.muni.cz

## Abstract

### Background

Specific toxic compounds, such as polycyclic aromatic hydrocarbons (PAHs) and their metabolites, may affect the inhalation injury (INHI) grade, patients' status, and prognosis for recovery. This pilot prospective study aimed to: i) evaluate the suitability of bronchoalveolar lavage (BAL) for determination of PAHs in the LRT and of urine for determination of hydroxylated metabolites (OH-PAHs) in patients with INHI, ii) describe the dynamic changes in the levels of these toxic compounds, and iii) correlate these findings with clinical variables of the patients with INHI.

### Methods

The BAL and urine samples from 10 patients with INHI were obtained on Days 1, 3, 5, 7, and 14 of hospitalization, if possible, and PAHs (BAL) and OH-PAHs (urine) were analyzed using chromatographic methods (GC-MS and HPLC).

### Results

Concentrations of analyzed PAHs were in most cases and time points below the limit of quantification in BAL samples. Nine OH-PAHs were detected in the urine samples; however, their concentrations sharply decreased within the first three days of the hospitalization. On Day 14, the total amount of OH-PAHs in urine was higher in surviving patients with High-grade INHI ($\geq$3) than in those with Low-grade INHI ($<$3, p = 0.032). Finally, a significant

**Data Availability Statement:** All relevant data are within the manuscript and its Supporting information files.

**Funding:** The study was supported by project provided by University Hospital Brno, Ministry of Health Czech Republic – RVO (FNBr, 65269705). This publication was supported from the European Union's Horizon 2020 Research and Innovation Programme under grant agreement No 857560. This publication reflects only the author's view and the European Commission is not responsible for any use that may be made of the information it contains. Authors also thank to Research Infrastructure RECETOX RI (No LM2023069) financed by the Ministry of Education, Youth and Sports for supportive background. The funders had no role in study design, data collection and analysis, decision to publish, or preparation of the manuscript".

**Competing interests:** The authors have declared that no competing interests exist.

correlation between certain OH-PAHs and clinical variables (AST/ALT, TBSA, ABSI) from Day 1 of the hospitalization was observed ($p < 0.05$).

## Conclusions

BAL samples are not suitable for the analysis of PAHs. However, the OH-PAHs levels in urine can be measured reliably and were correlated with several clinical variables. Moreover, High-grade INHI was associated with higher total concentrations of OH-PAHs in urine.

## 1. Introduction

Inhalation injury (INHI) is defined as an acute airway injury and mucosal damage caused by inhalation of hot steam and/or toxic compounds [1]. Besides increasing the mortality and morbidity of patients with burns [2–5], INHI can be life-threatening on itself, especially if secondary complications such as bacterial and/or fungal infections occur [6]. The death rates in patients with INHI range from 1.3% to 58.2% [6,7]. INHI is sometimes very difficult to diagnose—the diagnosis is mostly performed on the basis of combined clinical presentation of the patient and bronchoscopy [8,9]. At present, however, there is no standardized procedure for INHI diagnosis and grading [3,10,11].

Hot air usually causes INHI only to the airway structure above the carina [4]. The hot air is cooled down during its passage through the upper respiratory tract (URT); the lower respiratory tract (LRT) is, therefore, affected rather by toxins and chemicals than by heat [4,6]. While the URT of patients with INHI is cleaned from solid compounds after hospitalization, inhaled toxins and their metabolites may persist in the LRT of these patients for a longer time. For this reason, where inhaled toxins are concerned, the lungs are the most vulnerable location of the respiratory tract. However, the local effects of toxins in the lungs and the time of their excretion are not fully clear.

These toxins comprise primarily the main products of combustion such as carbon monoxide [12], ammonia [13], hydrogen fluoride [14,15], hydrogen sulfide, cyanide, etc. [16]; in addition, more complex chemical compounds, such as dangerous polycyclic aromatic hydrocarbons (PAHs) [17] or polychlorinated biphenyls (PCBs), may also enter the LRT and are excreted from the human body by urine in the form of metabolites [18]. Although many studies have investigated the risk posed by human exposure to PAHs, no study investigating PAHs in the LRT of patients with INHI is available to this date, and only limited data can be found on hydroxylated metabolites of PAHs (OH-PAHs) levels in urine in such patients. The level and duration of exposure to these toxic compounds can strongly influence the INHI severity as well as treatment and its success [19]; this aspect has, however, not been sufficiently studied so far.

We hypothesized that the exposure to specific toxic compounds, together with the human body's ability to remove them, is associated with the grade of INHI, patient's health status, and prognosis. Therefore, we assume that the analysis of toxic compounds, such as specific toxins and/or their metabolites, can be used as an early predictor of the disease outcome. If this is confirmed, monitoring of such compounds can be beneficial in the management of personalized therapy in INHI patients.

This pilot study aimed to determine matrices and analytes suitable for further research in patients with INHI, namely to: i) evaluate the suitability of BAL for determining PAHs in the LRT and suitability of urine for determining hydroxylated metabolites (OH-PAHs) in patients

with INHI, ii) describe the dynamic changes in the levels of these toxic compounds throughout hospitalization in LRT and in urine, and iii) correlate these findings with clinical variables of the patients with INHI at the beginning of the hospitalization.

## 2. Materials and methods

### 2.1 Study design and clinical examination

This prospective monocentric study focused on monitoring of the exposure to toxic substances and their metabolites in two body fluids (bronchoalveolar lavage and urine) of patients with INHI (n = 10) throughout their hospitalization at the Department of Burns and Plastic Surgery, University Hospital Brno, Czech Republic, from 19/05/2020 to 06/05/2022. The study was designed as a case series study due to the rarity of the injury. The patients were sampled at five time points throughout their hospitalization (Days 1, 3, 5, 7, and 14 of hospitalization).

The study was approved by the Ethics Committee of the University Hospital Brno, Czech Republic (No. 04-120220/EK, date 12/FEB/2020). Written informed consent was obtained from all participants or their closest relatives prior to their inclusion in the study. All procedures were in accordance with the ethical standards of the institutional and/or national research committee and with the 1964 Helsinki Declaration as amended.

The inclusion criteria were: INHI diagnosis (confirmed by bronchoscopy and laryngoscopy evaluation with complete anamnestic data), age 18–70 years, Czech or Slovak origin, need for intubation and mechanical ventilation for at least 72 hours, signed written informed consent. The exclusion criteria were: inability to follow study procedures, survival prognosis of less than 48 hours, enrolment in any other investigational trial within 4 weeks prior to screening, relationship to the clinical study site or investigator, prisoners and incarcerated persons, chronic use of systemic steroids or other immunosuppressants, and delayed transfer to the burn center (> 48 hours).

In the patients included in the pilot study, the INHI grade was evaluated [9], the extent of the burned surface area (TBSA) was determined, and the Abbreviated Burn Severity Index (ABSI) [20] was calculated. Furthermore, basic biochemical parameters (C-reactive protein—CRP, Procalcitonin, Blood Urea Nitrogen, Creatinine, Glucose, Lactate, Natrium, Potassium, Chlorides, Phosphate, Magnesium, Bilirubin, Albumin, Alanine Aminotransferase, Aspartate Aminotransferase) were determined in these patients throughout hospitalization. Standard microbiological examinations were also carried out in different compartments with an emphasis on the cultivation of material from the LRT.

### 2.2 Sample collection and preparation

The samples of bronchoalveolar lavage (BAL) and urine were collected from patients with INHI on Days 1, 3, 5, 7, and 14 of hospitalization following the standard operation procedure. The BALs were collected during bronchoscopy from 5 sites in the lungs (one from each lobe), while urine was collected from the urinary catheter.

The samples were aliquoted as follows: BALs were pooled from all lung sites to obtain 1 mL of sample on each of Days 3, 5, and 7 of hospitalization. On Days 1 and 14 of hospitalization, the BALs from individual lobes were pooled for each lung separately. The BALs from the right lung constituted 600 μL of sample (three lobes, 200 μL each) and from the left lung (two lobes), 400 μL was collected. 500 μL of urine sample from each patient at each time point was used for toxicological analysis and 200 μL for determination of creatinine.

## 2.3 Analysis of polycyclic aromatic hydrocarbons

Analytical standards of native PAHs (LGC, UK), deuterated PAHs (Supelco, USA), native nitro- and oxy-PAHs (Accustandard, USA and Chiron, Norway), deuterated nitro- and oxy-PAHs (Chiron, Norway), native OH-PAHs (Neochema, Germany and Sigma-Aldrich, USA), $^{13}$C OH-PAHs (Cambridge Isotope Laboratories, USA and Toronto Research Chemicals, Canada) were used for quantification of samples. Dichloromethane (DCM) and *n*-hexane (pesticide residue grade) were purchased from JT Baker (Avantor, USA); silica, formic acid (98–100%), β-Glucuronidase from Escherichia coli (type IX-A, 2MU) was purchased from Merck (Germany); sodium acetate, ammonium fluoride, acetic acid, dimethylsulphoxide (all p.a.) were purchased from Sigma-Aldrich (USA); methanol (LC-MS grade) was purchased from Biosolve (France). The detailed list of used chemicals is provided in S1 File.

The composition and amounts of PAHs and their derivatives such as nitro-PAHs and oxy-PAHs were measured in BAL samples. These samples were spiked with deuterated PAHs, deuterated nitro- and oxy-PAHs, and extracted using liquid-liquid extraction. The samples were extracted using 10 mL of a *n*—hexane-dichloromethane (4:1) mixture for 5 minutes using an orbital shaker. The extraction was repeated three times and the extracts pooled for further clean-up.

The extract was cleaned on a silica column (3 g of silica, 0.063–0.200 mm, activated at 150˚C for 12 hours, 10% deactivated with MiliQ water) with 1 g $Na_2SO_4$. The pooled BAL aliquotes (see) was loaded and eluted with 5 mL *n*-hexane followed by 40 mL DCM. All chemicals and sorbents used for analyses were at minimum of pesticide residue grade. The eluate volume was then reduced by a stream of nitrogen in a SuperVap (FMS, USA) concentrator unit and transferred into a vial. Terphenyl and $^{13}$C PCB 95 were added as syringe standards, the final sample volume was 200 µL. PAHs, nitro-PAHs, and oxy-PAHs were subsequently determined using the methods below.

PAHs were analyzed on 8890A GC (gas chromatography, Agilent, USA) equipped with a 60 m × 0.25 mm × 0.25 µm Rxi-5Sil-MS column (Restek, USA), coupled to a triple quadrupole 7000D MS (Agilent, USA). The temperature program for the GC oven started at 80˚C (2 min hold), then continued with 15˚C/min to 180˚C (no hold), and lastly 5˚C/min to 310˚C (20 min hold). The inlet temperature was 280˚C. The injection volume was 1 µL in pulsed-splitless mode. The carrier gas was helium with a flow rate of 1.5 mL/min. The temperature of the GC-MS transfer line was 310˚C. The ion source was heated to 320˚C. The mass spectrometer was operating in selected ion monitoring (SIM) mode. Compound quantification was done in the MassHunter Workstation 10.1 software.

Nitro- and oxy-PAHs were analyzed by atmospheric pressure chromatography-tandem mass spectrometry (APGC-MS/MS) on a Waters Xevo TQ-S MS (Waters, UK) coupled to Agilent 7890 GC (Agilent, USA). The MS was operated under dry source conditions in multiple reactions monitoring mode (MRM). The GC was fitted with a 30 m × 0.25 mm × 0.25 um Rxi-5Sil MS column (Restek, USA). Splitless injection at 270˚C was used. Helium was used as a carrier gas at a constant flow of 1.5 mL/min. The oven temperature program started at 90˚C (1 min hold), then gradients of 40˚C/min to 150˚C and 5˚C/min to 250˚C (5 min hold) were applied, followed, finally, by a gradient of 10˚C/min to 320˚C (5 min hold).

In urine samples, only OH-PAHs were measured using liquid chromatography with tandem mass spectrometry (LC-MS/MS). Urine samples were extracted using a modified CDC protocol 6705.02 [21]. Briefly, the samples were thawed and homogenized. 500 µL of the sample was mixed with 500 µL of sodium acetate buffer (0.1M, pH 6) containing β-glucuronidase (*Escherichia coli* Type IX-A, 500U) and internal standards ($^{13}$C and Deuterium labeled). The mixture was incubated for 120 min at 55˚C. Analytes were extracted using solid phase

extraction SPE 96-well plate cartridges (Water Oasis HLB, 60mg, 30μm). All chemicals and solvents were at least of LC-MS/p.a. grade. Native and isotopically labeled standards were purchased from Neochema (Bodenheim, Germany) and Cambridge Isotope Laboratories (Tewksbury, MA, USA). OH-PAHs were analyzed on the Agilent 1200 series liquid chromatography (HPLC) system (Agilent Technologies, Waldbronn, Germany). Chromatographic separation was accomplished using a Waters Acquity BEH C-18 analytical column (100 x 2.1 mm, 1.7 μm particle size) maintained at 30˚C and equipped with the Acquity BEH C-18 VanGuard pre-column (Waters, Milford, MA, USA). The mobile phases for the gradient separation of the analytes were 0.1 mM ammonium fluoride in MilliQ water (component A) and 0.1 mM ammonium fluoride in methanol (component B). The flow rate was 0.3 mL/min, and the injection volume was 5 μL. Analytes were detected by a tandem mass spectrometer AB Sciex QTrap 5500 operating in negative electrospray ionization mode at 450˚C with $N_2$ as a nebulizer gas and a capillary voltage of -4 kV (SCIEX, Concord, ON, Canada) as published previously [22].

### 2.4 Quality assurance and quality control (QA/QC)

Procedural blanks were processed in the same way the samples were. Recoveries of the labeled compounds are provided (S1 File) and data are not recovery-corrected. For nitro-PAHs and oxy-PAHs, the limit of quantification (LOQ) was calculated for each analyte and sample as the concentration corresponding to the signal-to-noise (S/N) ratio of 10. For PAHs, LOQ values were calculated per batch using calibration standards solutions as concentrations corresponding to S/N of 10 and are detailed in S1 File.

Deuterium- and $^{13}$C-labeled OH-PAHs isotope dilution method was used for the quantification of the analytes. The linear quantification range was 0.01–100 μg/Ls urine, with limits of quantification from 0.018 to 0.2 μg/L urine for the respective analytes (MQL, calculated as 10× the standard deviation (SD) of the blank sample set concentration). With each analytical batch, a set of blank samples (matrix) and certified reference material samples (NIST CRM® 3672 and 3673) were analyzed. Quality control was ensured by participation in the HBM4EU quality assurance ICI-EQAS program [23] and OSEQAS quality assessment scheme organized by Centre de Toxicologie du Québec (INSPQ), successfully completed since 2018 [24].

### 2.5 Creatinine analysis

Creatinine (mmol/L) was measured in urine samples on Cobas Integra 400 (Roche Diagnostics).

### 2.6 Statistical analysis

Patients were divided into two groups according to the INHI grade. The patients with INHI grade lower than 3 (grade < 3) were designated as Low-grade INHI group, patients with an INHI grade of 3 or higher (grade ≥ 3) were designated as High-grade INHI group.

Basic statistics were used to describe the measured parameters, i.e., the median and mean as indicators of the center and the standard deviation together with the interquartile range (IQR) as indicators of variability. The kinetics of individual metabolites were shown using graphs with marked measurement values on individual days of hospitalization. Limit values were also shown in the graphs (where existing). Due to the small number of measured values and large skewness, the correlation was measured using the non-parametric Spearman correlation coefficient.

The non-parametric Wilcoxon rank sum test with continuity correction was used to test the differences between the two groups classified according to the INHI grade. For Patient 4, who died during hospitalization, the last detected value from Day 7 was used also for Day 14.

## 3. Results

All raw data for this manuscript are shown in S2 File.

### 3.1 Clinical characterization of patients

Ten patients were included in this study, with both High-grade INHI and Low-grade INHI containing five patients. Two patients from the High-grade INHI group died during the hospitalization. The patients generally suffered extensive burns on the body surface (median TBSA = 26%, IQR = 10.75–35.75% on Day 1 of hospitalization). AST/ALT ratio ranged between 0.641 and 5.583 (median = 2.094, IQ = 1.45–2.71; physiological range 0.205–1.197, see Fig 1a). All the patients had hypoproteinemia, the values of the total protein ranged between 28.6 and 75.2 g/L (median = 55.3 g/L, IQR = 48.93–63.33 g/L; physiological range 64-83 g/L, see Fig 1b). However, each patient had unique clinical characteristics because of the origin of the injury. See the detailed characterization of each patient in S3 File.

### 3.2 PAHs in bronchoalveolar lavage samples

In general, only trace amounts of the measured PAHs and their nitro- and oxy-derivatives (further only PAHs) were detected in BAL samples (< 0.1 ng/mL). The concentrations typically ranged between 0.003 and 3.8 ng/mL, with 1,4-naphthoquinone being the only exception —its median concentration was 130.5 ng/mL (IQR = 98.93–150.75 ng/mL). In total, only 19 out of the total 56 measured analytes (including PAHs and their derivatives; see Table 1) were above the LOQ at least once during the hospitalization in any of the patients.

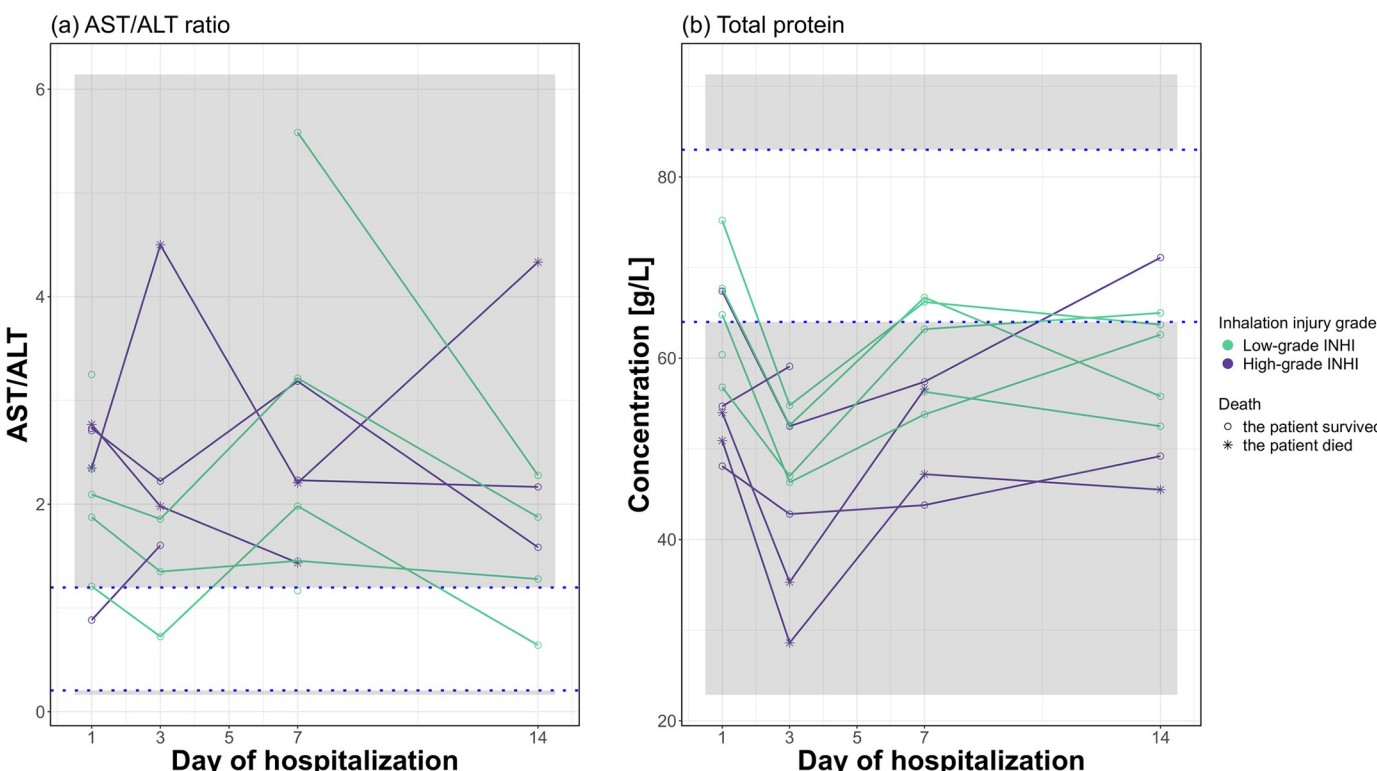

**Fig 1. Dynamic changes of the AST/ALT ratio (a) and the concentration of total protein (b) for ten patients with inhalation injury (INHI).** The white strip shows the physiological range for the individual clinical variable.

**Table 1. Polycyclic aromatic hydrocarbons (PAHs) detected in bronchoalveolar lavage samples (combined results from the entire period).**

| PAHs [ng/mL] | N of samples in which the analyte was detected | N of patients with INHI in which the analyte was detected | Average concentration [ng/mL] | Median concentration [ng/mL] | Q1 [ng/mL] | Q3 [ng/mL] | SD [ng/mL] | IQR [ng/mL] |
|---|---|---|---|---|---|---|---|---|
| 1,4-naphthoquinone | 4 | 2 | 119.175 | 130.5 | 98.925 | 150.75 | 46.037 | 51.825 |
| 9,10-antraquinone | 15 | 6 | 3.801 | 3.312 | 2.645 | 4.730 | 1.599 | 2.085 |
| 6H-benzo[c]chromen-6-one | 8 | 4 | 1.908 | 1.828 | 1.398 | 2.225 | 0.639 | 0.827 |
| retene | 3 | 1 | 0.303 | 0.350 | 0.248 | 0.382 | 0.140 | 0.134 |
| benzo[g,h,i]fluoranthene | 2 | 1 | 0.259 | 0.259 | 0.231 | 0.288 | 0.081 | 0.057 |
| benzo[g,h,i]perylene | 3 | 2 | 0.212 | 0.209 | 0.193 | 0.230 | 0.037 | 0.037 |
| chrysene | 7 | 3 | 0.214 | 0.191 | 0.189 | 0.242 | 0.093 | 0.053 |
| 3-nitroacenaphthene | 1 | 1 | 0.101 | 0.101 | 0.101 | 0.101 | NA | NA |
| triphenylene | 5 | 3 | 0.120 | 0.091 | 0.055 | 0.135 | 0.1 | 0.080 |
| benzo[a]anthracene | 8 | 3 | 0.118 | 0.082 | 0.071 | 0.141 | 0.093 | 0.071 |
| benz[a]anthracene-7,12-dione | 1 | 1 | 0.051 | 0.051 | 0.051 | 0.051 | NA | NA |
| 1-nitropyrene | 9 | 5 | 0.046 | 0.040 | 0.038 | 0.061 | 0.014 | 0.023 |
| 2-nitrofluorene | 2 | 1 | 0.034 | 0.034 | 0.029 | 0.038 | 0.012 | 0.009 |
| 3-nitrophenantrene | 7 | 5 | 0.036 | 0.029 | 0.026 | 0.044 | 0.016 | 0.018 |
| 6-nitrochrysene | 1 | 1 | 0.029 | 0.029 | 0.029 | 0.029 | NA | NA |
| 7-nitrobenzo[a]anthracene | 5 | 2 | 0.018 | 0.018 | 0.016 | 0.020 | 0.007 | 0.005 |
| 5,12-naphthacenequinone | 1 | 1 | 0.016 | 0.016 | 0.016 | 0.016 | NA | NA |
| 9-nitrophenantrene | 1 | 1 | 0.004 | 0.004 | 0.004 | 0.004 | NA | NA |
| 1,3-dinitropyrene | 1 | 1 | 0.003 | 0.003 | 0.003 | 0.003 | NA | NA |

The table includes the data from 10 patients with inhalation injury and 5 sampling days; IQR, interquartile range; SD, standard deviation.

The PAHs in BAL samples were detected mostly in the samples collected on Day 1 of the hospitalization. In the following days, the concentrations of PAHs were detected also at other time points in some patients; however, as these concentrations were still close to the limit of detection, no meaningful conclusions on the PAH removal rate in the lungs could be made on the basis of BAL samples. Similarly, it was not possible to compare findings from the right and left lungs due to the lack of analytes in the BAL samples.

### 3.3 Concentration changes of OH-PAHs in urine throughout hospitalization

The originally high concentrations of some OH-PAHs decreased throughout hospitalization. The concentrations of OH-PAHs were higher in the High-grade INHI group than in the Low-grade INHI group (see Fig 2). The largest declines in the concentrations were observed, as a rule, over the first three days of hospitalization; the concentration of OH-PAHs decreased to concentrations around the minimum over this time.

Further, the trend of decreasing OH-PAHs concentrations in urine samples throughout the hospitalization was observed in the case of 2-hydroxynaphthalene (2-OH-Naph), 2-hydroxy-fluorene (2—OH-Fluo), 3-hydroxyfluorene (3-OH-Fluo), 1-hydroxyphenanthrene (1—OH—Phen), 4—hydroxyphenanthrene (4-OH-Phen), 9-hydroxyphenanthrene (9—OH-Phen), 1—hydroxypyrene (1—OH—Pyr), 2/3-hydroxyphenanthrene (2/3-OH-Phen), and 1—hydroxy-naphthalene (1-OH-Naph). These nine listed OH-PAHs form the sum of OH-PAHs.

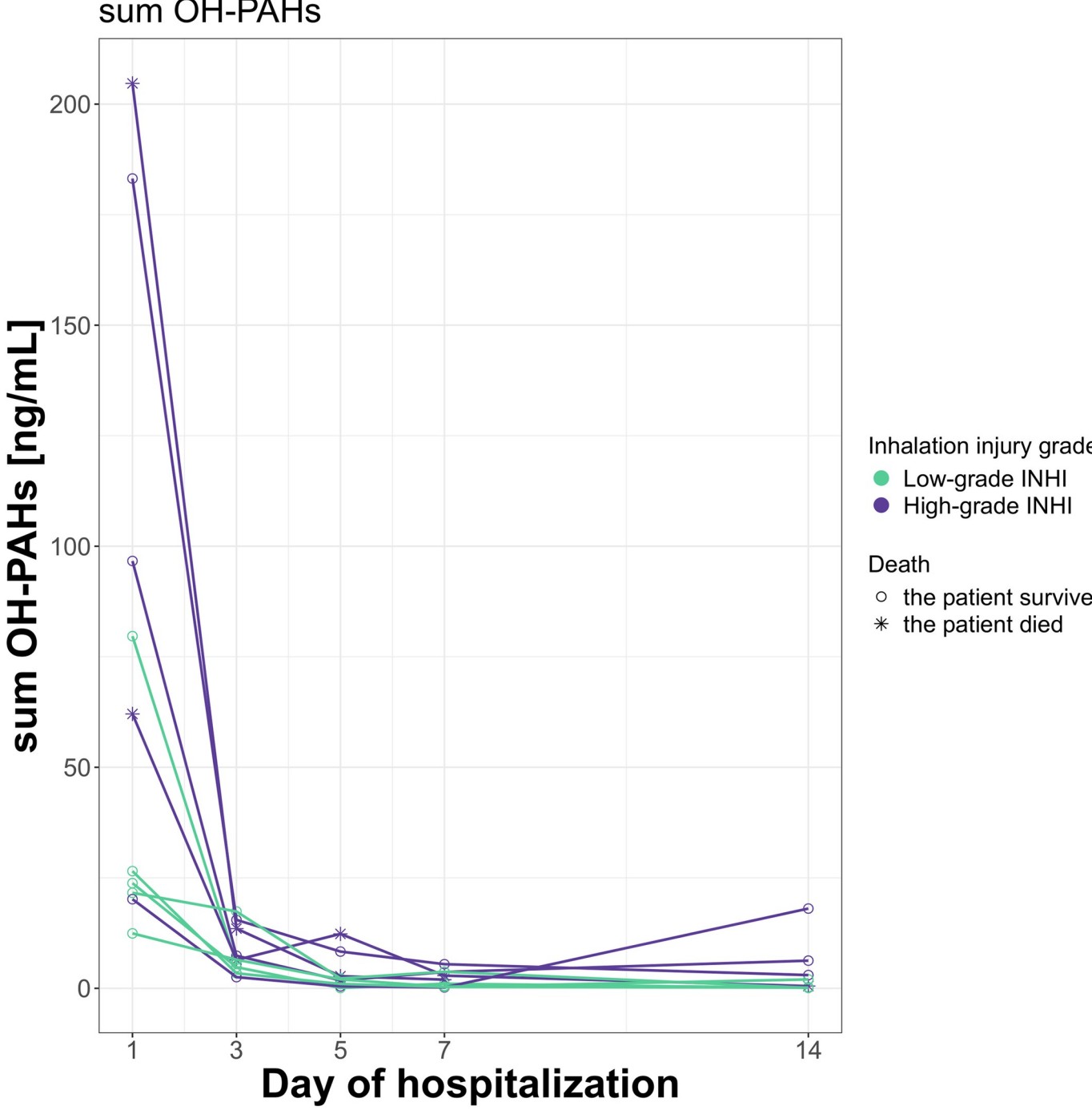

**Fig 2. The kinetic graph of the sum of metabolites of polycyclic aromatic hydrocarbons (OH-PAHs) in the urine samples from ten patients with inhalation injury (INHI).**

The correlations among nine individual OH-PAHs are shown in Fig 3, with significance indicated where applicable. All correlations were positive. The 4—OH—Phen and 2/3-OH-Phen correlated with all other eight detected OH-PAHs; 3-OH-Fluo and 1-OH-Naph were associated with seven OH—PAHs. The 3-OH-Fluo positively correlated with four other

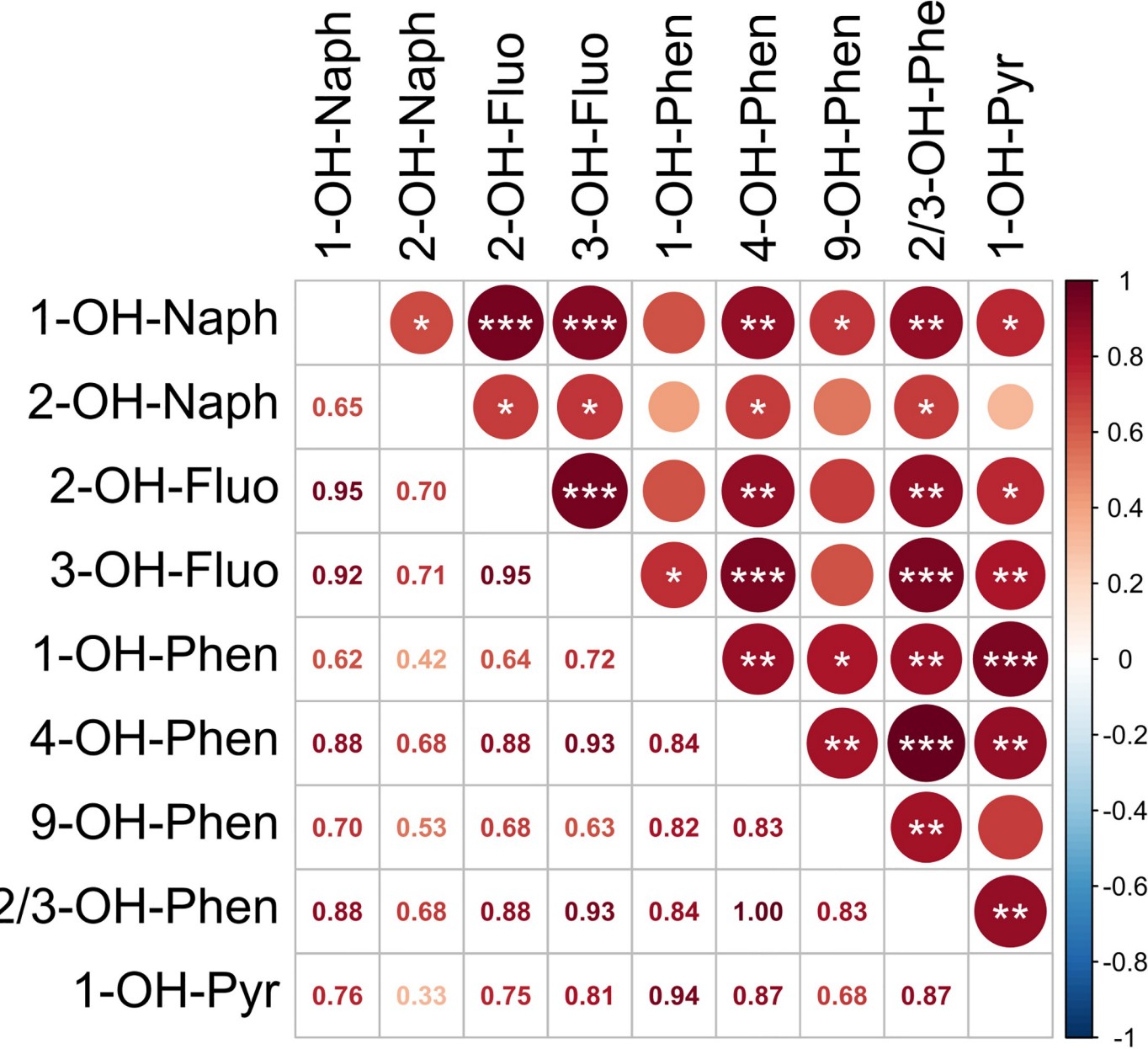

**Fig 3. Correlation plot between metabolites of polycyclic aromatic hydrocarbons (OH-PAHs, in ng/mL).** The Spearman´s correlation coefficients are below the diagonal. Statistical significance is highlighted (above the diagonal): ***p < 0.001, **p < 0.01, *p < 0.05.

OH—PAHs: 1—OH—Naph, 2-OH-Fluo, 4—OH-Phen, and 2/3—OH—Phen (p < 0.001). 1-OH-Naph showed strong correlations with both 2-OH-Fluo and 3-OH-Fluo (p < 0.001). Changes in the concentrations of individual OH—PAHs for each patient are provided in S3 File.

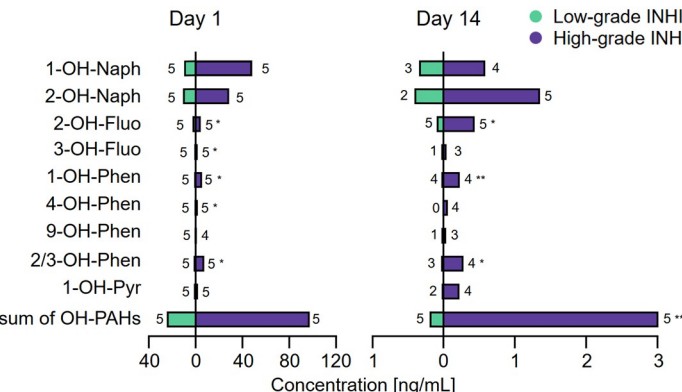

**Fig 4. Differences in the metabolites of polycyclic aromatic hydrocarbons (OH-PAHs) in urine groups with different inhalation injury (INHI) severity.** The differences are shown for Day 1 and Day 14 of hospitalization (**p < 0.05, *p < 0.1). The number of included patients is shown next to the bar-plot.

### 3.4 Differences in OH-PAHs concentrations between groups of patients with inhalation injury

Some differences in OH-PAHs concentrations in urine samples were found between the patients with Low-grade and High-grade INHI (Wilcoxon rank sum test; see Fig 4). In general, levels of OH-PAHs were higher in patients with High-grade INHI than in those with Low-grade INHI.

### 3.5 Correlation of OH-PAHs in urine samples with clinical parameters

A positive correlation was observed between some OH-PAHs in urine and clinical variables on Day 1 of hospitalization. Concentrations of certain OH-PAHs positively correlated with age, TBSA, ABSI score, total bilirubin, and AST/ALT ratio, and negatively with concentrations of potassium ions and albumin (see Fig 5).

## 4. Discussion

Our results showed that the measurement of OH-PAHs in urine has the potential to be used for the assessment of INHI progression and the determination of its grade. So far, no studies have investigated toxic compounds such as PAHs and their effect directly on INHI progression. Additionally, there is a lack of information about the toxic compounds in patients with INHI in general, even though it is likely that the toxic compounds can strongly influence the severity of the INHI [19].

Typically, sixteen PAHs listed by the United States Environmental Protection Agency (US EPA) are analyzed as priority PAHs, namely: naphthalene, acenaphthylene, acenaphthene, fluorene, anthracene, phenanthrene, fluoranthene, pyrene, chrysene, benz[a]anthracene, benzo[b]fluoranthene, benzo[k]fluoranthene, benzo[a]pyrene, indeno[1,2,3cd]pyrene, benzo[g,h,i]perylene, and dibenz[a,h]anthracene [25]. In our study, we targeted in total 56 PAHs (including priority pollutants and their derivatives) in BAL samples; however, only 19 of them were detected, of which only three (benzo[a[anthracene, benzo[g,h,i]perylene, and chrysene) are among the selected 16 PAHs. The concentrations of these PAHs were mostly in trace amounts, which is in line with the findings by Che et al. who analyzed 16 priority PAHs in

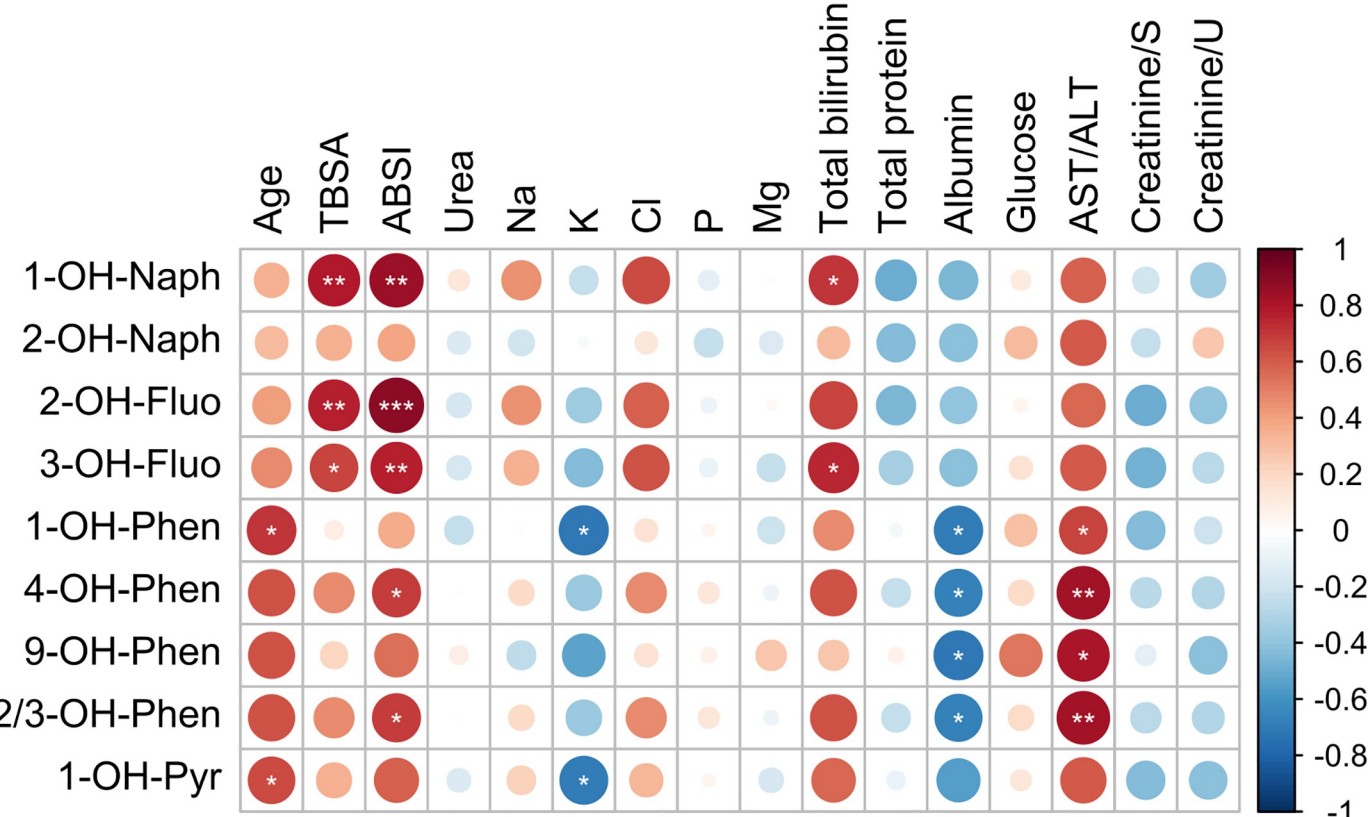

**Fig 5. Correlation between clinical variables and urine concentrations of metabolites of polycyclic aromatic hydrocarbons (OH-PAHs, in ng/mL).** The correlation is shown for Day 1 of hospitalization. Statistical significance: ***$p < 0.001$, **$p < 0.01$, *$p < 0.05$. Creatinine/S—creatinine in blood serum, creatinine/U—creatinine in urine.

BAL samples from patients with chronic obstructive pulmonary disease (COPD) [26]. Chrysene and benzo[a]anthracene were among PAHs detected in their study, similar to ours [26].

No analysis of PAHs in BAL from patients with INHI has been reported yet. Detection of the PAHs in the airways is difficult, most likely due to their lipophilic nature and, in effect, their poor extraction into the solution used for BAL. According to our results, the concentrations of PAHs in BAL were minimal, with 9,10-anthraquinone and 1,4-naphthoquinone being the most commonly detected PAHs. These two quinones are the most polar compounds of the molecules we analyzed, even though their water solubilities are still low. Therefore, it is no surprise that they were the most extracted molecules from the airways into physiological saline solution (the aqueous phase) during bronchoscopy. Moreover, the generally relatively low water-soluble molecules penetrate deeper into airways, thus causing injury in the distal airways and alveoli [27]. Even though the airways are damaged, the clearance of PAHs from the LRT, i.e., their transfer to circulation and their detoxification (the conversion of PAHs into metabolites and their transport into the urine) is probably relatively fast. This suggestion is supported by our results for urine samples, which show a high concentration of OH-PAHs as soon as the first day of hospitalization and a relatively rapid decrease over the first three days of hospitalization. The drop in the urine concentrations of PAHs and their metabolites between Days 1 and 3 is likely due to the relatively rapid clearance of the PAHs from the LRT by mucociliary clearance through the upper respiratory tract [28], thus limiting further intake of PAHs into the organism.

The mucociliary clearance is, therefore, an important factor. Under normal circumstances, approximately 20–30 mL of secretions are produced by the airways every day; it could be even more in patients with pulmonary disorders [29,30]. In general, the lungs have a fast clearance rate; the mucociliary clearance in the trachea is ~4-5 mm/min in healthy non-smoking adults. It is thought to be lower in patients with a lung disease such as COPD [31,32]. However, our results suggest that the lung clearance in patients with INHI might be similarly fast as under normal circumstances. In addition to the fast clearance of PAHs from the LRT, the metabolization of PAHs that entered the bloodstream (followed by their excretion by urine) is likely also relatively fast, although some accumulation into fatty tissues can be also observed. All these pathways probably contribute towards the observed steep initial drop in the urine concentrations of OH-PAHs that will be further discussed below.

Nevertheless, considering the low concentrations of PAHs in BAL samples despite the assumed high initial load of PAHs (as witnessed also by the relatively high OH-PAHs initial concentrations in the urine), BALs appear to be an unsuitable matrix for the analysis of PAHs. This is especially given by the non-polar nature of PAHs that are poorly extractable into the environment of the highly polar physiological solution. Hence, the results of BAL analysis must be interpreted with caution, and in this paper, we present the results of BAL analyses but do not claim them to have clinical relevance.

In addition, we performed a correlation analysis of OH-PAHs in urine. This analysis was performed with the sole purpose of finding out whether any single PAH or oxy-PAH could serve as a predictor of a) exposure to other PAHs and b) clinical outcomes. This would allow analyzing only a single PAH instead of the entire range, which would be much easier from the perspective of establishing the analytical method in clinical practice (and, in effect, also much more economically favorable). From this perspective, the hydroxylated naphthalene or fluoranthene appear to be the most promising compounds applicable for this purpose—OH-naphthalene correlates well with almost all other analyzed PAHs and can be, therefore, used as a general predictor for PAH exposure. Where clinical variables are concerned, all three aforementioned compounds correlated very well with the ABSI score. In the past, Wen et al. analyzed OH-PAHs in patients with COPD [33]; however, in contrast to their data, we found a significant correlation between almost all OH—PAHs. This difference, however, is not such a surprise; while COPD is a chronic disease, INHI is an acute serious stage occurring often after extensive exposure to toxic compounds. Most commonly, we found positive correlations for 4—OH—Phen and 2/3-OH—Phen, which were correlated with all other OH-PAHs. It should be also noted that the strongest correlations were among i) 1-OH-Naph with 2-OH-Fluo and 3-OH-Fluo, ii) 2-OH—Fluo with 3-OH-Fluo, iii) 3—OH—Fluo with 4-OH-Phen and 2/3-OH—Phen, iv) 1-OH-Phen with 1—OH—Pyr, and v) 4-OH-Phen with 2/3-OH-Phen.

Wen et al. also reported that co-exposure to PAHs correlates with a higher risk of COPD [33], which has some similarities with our results in patients with INHI. We found out that the presence of 2—OH—Fluo, 1-OH-Phen, and 2/3-OH-Phen in urine correlated with a High-grade INHI ($\geq$ 3) in the later days of hospitalization. Moreover, a significant difference in the sum of OH-PAHs between patients with Low-grade ($<$ 3) and High-grade ($\geq$ 3) INHI was observed. Two patients with High-grade INHI and high concentrations of the total amount of OH—PAHs died during the hospitalization. Thus, we suggest that the total concentration of OH—PAHs from urine has a potential to be used as a prognostic marker for patients with INHI.

This paper is the first to evaluate the correlations between OH-PAHs from urine and clinical variables in patients with INHI. We targeted the results from the first day of hospitalization to evaluate the prognostic value of the OH-PAHs concentrations on the assessment of risk associated with INHI. Our results showed a significant positive correlation between OH-

fluorenes (2-OH-Fluo and 3—OH-Fluo) as well as 1-OH-Naph with TBSA and ABSI score, with the strongest correlation being observed between 2-OH-Fluo and ABSI score. The ABSI score is one of the main approaches used for the determination of burn severity. The ABSI score estimates survival expectancy in a burn patient on the basis of negative prognostic factors including: i) sex, ii) age, iii) presence of inhalation injury, iv) presence of full-thickness burns, and v) the percentage of TBSA score [34]. Wang suggested a new approach for the assessment of burns based on physiological and biochemical variables, such as AST/ALT ratio and albumin concentration [35]. These clinical variables reflect the liver function (as liver malfunction is one of the major secondary complications in burn patients). Our results support their proposal; while a positive correlation between OH—phenanthrenes (1-OH-Phen, 4-OH-Phen, 9 —OH—Phen, and 2/3—OH—Phen) and AST/ALT ratio was found in patients with INHI, significant negative correlations of OH—phenanthrenes with albumin were detected. Our results suggest that the toxic compounds to which the patients were exposed might be considered negative prognostic factors when talking about INHI, with the occurrence of OH-fluorenes and OH—phenanthrenes being the most significant.

We believe that our research might have clinical implications in the future, allowing better assessment, risk stratification, and estimation of the development of secondary complications of INHI. However, our study has also several limitations. First of all, the number of patients in this study was small, which is a common limitation in the case of rare diseases. On the other hand, we performed prospective sampling of BAL and urine, which allowed us to study the dynamic changes in the analyzed toxic compounds over time. Another strength of our study lies in the robust, long-established methodology for the determination of PAHs and OH— PAHs in our core facility using GC-MS and HPLC, respectively [25].

## 5. Conclusions

Our results showed firstly that BAL samples from patients with INHI appear to be an unsuitable matrix for analysis of PAHs and secondly, more importantly, the hydroxylated metabolites of PAHs (OH—PAHs) are present in urine in sufficient amounts to be analyzed using chromatography connected to mass spectrometric detector. The concentration of OH-PAHs strongly correlates with some clinical variables, such as the AST/ALT ratio, TBSA, and ABSI scores. Further, total amounts of OH—PAHs in urine predicted the prognosis in patients with INHI. We believe that these findings could be implemented into the INHI diagnosis and grading management. Moreover, urine is easily obtainable, and the sampling is non-invasive; it is, therefore, the perfect matrix for the analysis of OH-PAHs in patients with INHI, which is a huge advantage of this method. Still, these results need to be confirmed in a larger cohort. Nevertheless, this pilot study successfully built the very basic pillars for involving toxic compounds in INHI management.

## Supporting information

**S1 Data. Raw clinical data of ten patients with inhalation injury during their hospitalization.** Raw data from PAHs, oxy-PAHs, nitro-PAHs analyses in pooled BAL samples of ten patients with inhalation injury during their hospitalization. Raw data from OH-PAHs analyses in urine samples of patients with inhalation injury during their hospitalization.
(XLSX)

**S1 File. PAHs information (SIM, RT, linearity, calibration range).** Nitro-PAHs information (MRM, RT, linearity, calibration range). Oxy-PAHs information (MRM, RT, linearity, calibration range). Method performance parameters (% recovery average, % RSD). List of used

chemicals and their purity. PAHs, oxy-PAHs, nitro-PAHs LOD and LOQ values in BAL samples. OH-PAHs LOD and LOQ values in urine samples.
(PDF)

**S2 File. Polycyclic aromatic hydrocarbons and their metabolites in bronchoalveolar lavage and urine samples from patients with inhalation injury throughout their hospitalization: a prospective pilot study.** The supplementary data contain a basic description of each of the ten patients included in our study: i) the characterization of the patients and their medical history throughout hospitalization with the inhalation injury, such as infection occurrence, ii) changes in clinical markers during hospitalization, and iii) changes in polycyclic aromatic hydrocarbons (PAHs) and their metabolites (OH-PAHs) during hospitalization.
(PDF)

## Acknowledgments

We thank Ludmila Sebejova from RECETOX CELSPAC Population Studies for the creatinine analysis in urine samples and Jakub Martiník from RECETOX Trace Analytical Laboratories for participating in PAH analyses.

## Author Contributions

**Conceptualization:** Bretislav Lipovy, Pavel Čupr, Petra Borilova Linhartova.

**Data curation:** Bretislav Lipovy, Pavel Čupr, Petra Borilova Linhartova.

**Formal analysis:** Katerina Vyklicka, Petr Gregor, Petra Borilova Linhartova.

**Funding acquisition:** Bretislav Lipovy, Petra Pribylova, Petra Borilova Linhartova.

**Investigation:** Katerina Vyklicka, Bretislav Lipovy, Filip Raska.

**Methodology:** Petr Kukucka, Jiri Kohoutek, Petra Pribylova.

**Project administration:** Bretislav Lipovy, Petra Borilova Linhartova.

**Supervision:** Bretislav Lipovy, Pavel Čupr, Petra Borilova Linhartova.

**Visualization:** Katerina Vyklicka, Petr Gregor, Bretislav Lipovy, Petra Borilova Linhartova.

**Writing – original draft:** Katerina Vyklicka, Petr Gregor, Bretislav Lipovy, Petra Borilova Linhartova.

**Writing – review & editing:** Filip Raska, Petr Kukucka, Jiri Kohoutek, Petra Pribylova, Pavel Čupr.

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
