## [Decision Letter · Decision Letter 0]

7 Jun 2024

PONE-D-24-10533Polycyclic aromatic hydrocarbons and their metabolites in bronchoalveolar lavage and urine samples from patients with inhalation injury throughout their hospitalization: a prospective pilot studyPLOS ONE

Dear Dr. Linhartova,

Thank you for submitting your manuscript to PLOS ONE. After careful consideration, we feel that it has merit but does not fully meet PLOS ONE’s publication criteria as it currently stands. Therefore, we invite you to submit a revised version of the manuscript that addresses the points raised during the review process. I strongly suggest that you take into account the reviewers' suggestions which are related to technical and experimental issues.

We look forward to receiving your revised manuscript.

Kind regards,

Giovanni Signore

Academic Editor

PLOS ONE

Journal Requirements:

3. Thank you for stating the following financial disclosure: "The study was supported by project provided by University Hospital Brno, Ministry of Health Czech Republic – RVO (FNBr, 65269705). This publication was supported from the European Union’s Horizon 2020 Research and Innovation Programme under grant agreement No 857560. This publication reflects only the author's view and the European Commission is not responsible for any use that may be made of the information it contains. Authors also thank to Research Infrastructure RECETOX RI (No LM2023069) financed by the Ministry of Education, Youth and Sports for supportive background. "

4. In the online submission form, you indicated that raw data will be available upon request from the corresponding author.

Reviewers' comments:

Reviewer's Responses to Questions

**Comments to the Author**

1. Is the manuscript technically sound, and do the data support the conclusions?

Reviewer #1: Yes

Reviewer #2: Yes

Reviewer #3: Partly

2. Has the statistical analysis been performed appropriately and rigorously? 

Reviewer #1: Yes

Reviewer #2: Yes

Reviewer #3: Yes

3. Have the authors made all data underlying the findings in their manuscript fully available?

Reviewer #1: Yes

Reviewer #2: Yes

Reviewer #3: No

4. Is the manuscript presented in an intelligible fashion and written in standard English?

Reviewer #1: No

Reviewer #2: Yes

Reviewer #3: Yes

5. Review Comments to the Author

Reviewer #1: Comments to the Authors:

- English writing needs further polish.

- Please avoid the keywords that are already in title of the MS.

- The section of "Introduction" is not well organized to reveal the importance of the work. It does not the sufficient solidarity. Therefore, the structure and logic of the introduction need to be modified.

- All reagents (purity, company, city, country) and all instruments (company, city, and country) must be included.

- Quality of the discussion section must be improved.

- The conclusions of your paper are especially important for this. Therefore, please try to sharpen this further. The optimal Conclusion should include:

* A summary of your key findings.

* A highlight of your hypothesis, new concepts, and innovations.

* A summary of key improvements compared to findings in the literature (provide a couple of references to indicate key improvements).

* Limitations of the study.

* Your vision for future work.

- For adding credibility to this work, add at least two representative chromatograms illustrating the separation of PAHS.

- The authors should include the following references to be added in the introduction section:

https://doi.org/10.1080/10406638.2023.2228453

https://doi.org/10.1080/15320383.2023.2215867

Reviewer #2: Presented by Vyklicka et al. manuscript titled "Polycyclic aromatic hydrocarbons and their metabolites in bronchoalveolar lavage and urine samples from patients with inhalation injury throughout their hospitalization: a prospective pilot study" covers the vital problem of early predictors of the diseases: PAHs and their derivatives hazard and presence in various matrices. Manuscript is interesting and novel. My specific comments:

Fig 1. Lack of units by title of the axix y (concentrationn), rather AST/ALT ratio (-). In gerenarl, the quality of Fig. i rather low.

l. 190-192: In total, only 19 out of the total 56 measured analytes (including PAHs and PAHs derivates; see Table 1) were above the LOQ at least once during the hospitalization in any of the patients.

what may be the explanation? may the extraction method affect the results?

line 195-196: he PAHs were detected mostly in the samples collected on Days 1 and 14 of the hospitalization.

how can it be explained?

204-205: Low-Grade of INHI (see Fig 2). The largest declines in the concentrations were observed, as a rule, over the first 3 days of hospitalization

how can it be explained?

l.290: but why such correlations are so important? what can be obtained/prediced from them?

conclusions: BAL samples from patients with INHI appear to be an unsuitable matrix for analysis of PAHs.

but nothing was mentioned on it in Discussion section,

how this conclusion was evaluated?

the authors stess the adventages of chromatographc methos, but the obejctives of the studies wans not establishing the novel method, no other methods for comparison were used.

Reviewer #3: The proposal of the manuscript is very interesting. However, more detailed information about the analytical methods applied is important. For example, this data could be added as Supplementary Material:

- ions used for quantification and qualification (and retention time) of analytes analyzed by GC-MS and LC-MS.

- more details about the materials used in the study such as solvents, reagents, chemicals, and standards, including manufacturers, purity, solutions preparation...

- Page 6 (line 114): What is the sample amount used for extraction?

- Page 7 (line 137): "using a modified CDC protocol 6705.02 (19)": add a brief description of the protocol.

- Page 7 (line 143): " Analyte detection was achieved by means of tandem mass spectrometer AB Sciex QTrap 5500 operating in negative electrospray ionization mode...": add information about the colision energy used to obtain the quantification and qualification ions.

- Data about the performance characteristics of the analytical methods is very important such as limit of detection, limit of quantification, linearity, recovery and precision.

6. PLOS authors have the option to publish the peer review history of their article (what does this mean?). If published, this will include your full peer review and any attached files.

Reviewer #1: No

Reviewer #2: No

Reviewer #3: No

---

## [Author Response · Author response to Decision Letter 0]

27 Jun 2024

Response to Reviewers

PONE-D-24-10533

Polycyclic aromatic hydrocarbons and their metabolites in bronchoalveolar lavage and urine samples from patients with inhalation injury throughout their hospitalization: a prospective pilot study.

Dear reviewers,

Thank you for the opportunity to revise our paper. We appreciate your insightful comments, suggestions, and critiques. We have incorporated these in the revised version of the manuscript (all changes are highlighted by yellow) or provided explanations where necessary. Below, we address all points raised.

We hope that you will be happy with the revised version of the manuscript as well as with the responses to your comments.

Thank you very much for your help,

Best regards,

Authors

Reviewer #1: 

Comment 1: English writing needs further polish.

Answer 1: The paper was sent for proofreading to a professional proofreading agency.

Comment 2: Please avoid the keywords that are already in title of the MS.

Answer 2: Thank you for the heads-up. It was corrected (see line 39).

Keywords: toxicokinetics, respiratory tract damage, urine biomarkers, toxic compounds

Comment 3: The section of "Introduction" is not well organized to reveal the importance of the work. It does not the sufficient solidarity. Therefore, the structure and logic of the introduction need to be modified.

Answer 3: Thank you for the suggestion. You were right that some of the paragraphs did not easily follow one another. We have restructured the Introduction to give the text better flow now (see lines 41-64). 

Comment 4: All reagents (purity, company, city, country) and all instruments (company, city, and country) must be included.

Answer 4: Thank you for the comment, we added this information to Chapter 2.3 and to the Supplementary Tables.

Comment 5: Quality of the discussion section must be improved.

Answer 5: Thank you, the quality of Discussion was improved per your suggestions and suggestions by other reviewers (see lines 288-328).

Comment 6: The conclusions of your paper are especially important for this. Therefore, please try to sharpen this further. The optimal Conclusion should include:

* A summary of your key findings.

* A highlight of your hypothesis, new concepts, and innovations.

* A summary of key improvements compared to findings in the literature (provide a couple of references to indicate key improvements).

* Limitations of the study.

* Your vision for future work.

Answer 6: Thank you for your suggestions, conclusions were amended (see lines 370-380). 

Comment 7: For adding credibility to this work, add at least two representative chromatograms illustrating the separation of PAHS.

Answer 7: Thank you for your comment. We attach here an example of a specific chromatogram of the separation of 2 groups of PAHs compounds (benzo(a)anthracene, triphenylene, chrysene and benzo(b/j/k)fluoranthene) on 60m Rxi-5Sil-MS column. However, procedures, methodologies and QA/QC are adequately described in the work.

Comment 8: The authors should include the following references to be added in the introduction section:

https://doi.org/10.1080/10406638.2023.2228453

https://doi.org/10.1080/15320383.2023.2215867

Answer 8: Thank you for the citation suggestions, we have carefully considered their appropriate placement in our article, but the cited citations are only marginally related to the topic of our work. We have managed to include one of the suggested studies, though (see lines 59-60).

Reviewer #2: 

Comment 1: Presented by Vyklicka et al. manuscript titled "Polycyclic aromatic hydrocarbons and their metabolites in bronchoalveolar lavage and urine samples from patients with inhalation injury throughout their hospitalization: a prospective pilot study" covers the vital problem of early predictors of the diseases: PAHs and their derivatives hazard and presence in various matrices. Manuscript is interesting and novel. 

Answer 1: We thank the reviewer for the time taken to review our paper and for positive evaluation.

Comment 2: Fig 1. Lack of units by title of the axix y (concentrationn), rather AST/ALT ratio (-). In gerenarl, the quality of Fig. i rather low. 

Answer 2: Thank you for your comment, we added the axis description in the Figure 1.a. As far as the low quality of the figures is concerned, this is just due to the pdf-generating procedure in the journal; the submitted figures are of sufficient quality.

Fig 1. Dynamic changes of the AST/ALT ratio (a) and the concentration of total protein (b) for ten patients with inhalation injury (INHI). The white strip shows the physiological range for the individual clinical variable. 

Comment 3: l. 190-192: In total, only 19 out of the total 56 measured analytes (including PAHs and PAHs derivates; see Table 1) were above the LOQ at least once during the hospitalization in any of the patients. What may be the explanation? may the extraction method affect the results?

Answer 3: This is a correct point. It is indeed likely given by the combination of the low amount of sample and poor solubility of PAHs/PAH derivatives in the physiological solution used for the alveolar lavage, as discussed in the text at lines 288-290.

Comment 4: line 195-196: The PAHs were detected mostly in the samples collected on Days 1 and 14 of the hospitalization. how can it be explained?

Answer 4: The fact that the highest concentrations were found in most patients on Day 1 is not surprising as on that day, the amounts of PAHs in the lower respiratory tract logically had to be the highest. Actually, drawing attention to Day 14 in the text is probably misleading – in other patients, PAHs/their derivatives were detected on other days as well but we have not stated that explicitly. In view of this, emphasizing just Days 1 and 14 was, therefore, incorrect. Please note also that the detected concentrations were (where detected) relatively close to the limit of detection anyway. For these reasons, we have replaced the formulation “Day 1 and 14” with “during hospitalization”. In any case, our study demonstrated that BAL samples are rather unsuitable for this type of analysis and rather highlight the relatively good yield in the urine samples.

The following was added to the Results section (see lines 224-229):

“The PAHs in BAL samples were detected mostly in the samples collected on Day 1 of the hospitalization. In the following days, the concentrations of PAHs were detected also at other time points in some patients; however, as these concentrations were still close to the limit of detection, no meaningful conclusions on the PAH removal rate in the lungs could be made on the basis of BAL samples. Similarly, it was not possible to compare findings from the right and left lungs due to the lack of analytes in the BAL samples.” 

Comment 5: 204-205: Low-Grade of INHI (see Fig 2). The largest declines in the concentrations were observed, as a rule, over the first 3 days of hospitalization. how can it be explained?

Answer 5: We have added the following explanation to the text (see lines 300-313):

“The drop in the urine concentrations of PAHs and their metabolites between Days 1 and 3 is likely due to the relatively rapid clearance of the PAHs from the LRT by mucociliary clearance through the upper respiratory tract [29], thus limiting further intake of PAHs into the organism. 

The mucociliary clearance is, therefore, an important factor. Under normal circumstances, approximately 20-30 mL of secretions are produced by the airways every day; it could be even more in patients with pulmonary disorders [30,31]. In general, the lungs have a fast clearance rate; the mucociliary clearance in the trachea is ~4 5 mm/min in healthy non smoking adults. It is thought to be lower in patients with a lung disease such as COPD [32,33]. However, our results suggest that the lung clearance in patients with INHI might be similarly fast as under normal circumstances. In addition to the fast clearance of PAHs from the LRT, the metabolization of PAHs that entered the bloodstream (followed by their excretion by urine) is likely also relatively fast, although some accumulation into fatty tissues can be also observed. All these pathways probably contribute towards the observed steep initial drop in the urine concentrations of OH-PAHs that will be further discussed below.”

Comment 6: l.290: but why such correlations are so important? what can be obtained/prediced from them?

Answer 6: You are right, this was not sufficiently explained in the manuscript. We have added the following to the Discussion (see lines 320-328):

“In addition, we performed a correlation analysis of OH-PAHs in urine. This analysis was performed with the sole purpose of finding out whether any single PAH or oxy-PAH could serve as a predictor of a) exposure to other PAHs and b) clinical outcomes. This would allow analyzing only a single PAH instead of the entire range, which would be much easier from the perspective of establishing the analytical method in clinical practice (and, in effect, also much more economically favorable). From this perspective, the hydroxylated naphthalene or fluoranthene appear to be the most promising compounds applicable for this purpose – OH-naphthalene correlates well with almost all other analyzed PAHs and can be, therefore, used as a general predictor for PAH exposure. Where clinical variables are concerned, all three aforementioned compounds correlated very well with the ABSI score.”

Comment 7: conclusions: BAL samples from patients with INHI appear to be an unsuitable matrix for analysis of PAHs. but nothing was mentioned on it in Discussion section, how this conclusion was evaluated?

the authors stess the adventages of chromatographc methos, but the obejctives of the studies wans not establishing the novel method, no other methods for comparison were used.

Answer 7: We have added the following paragraph into the Discussion (see lines 314-319):

“Nevertheless, considering the low concentrations of PAHs in BAL samples despite the assumed high initial load of PAHs (as witnessed also by the relatively high OH-PAHs initial concentrations in the urine), BALs appear to be an unsuitable matrix for the analysis of PAHs. This is especially given by the non-polar nature of PAHs that are poorly extractable into the environment of the highly polar physiological solution. Hence, the results of BAL analysis must be interpreted with caution, and in this paper, we present the results of BAL analyses but do not claim them to have clinical relevance.”

We do not highlight chromatography as the best method as (as you correctly pointed out) we have not compared it with any other method. The only mention of chromatography in Conclusions is that urine in these patients contains enough OH-PAHs to be analyzed by chromatography, i.e, that this method can be used (but make no implications about its better or worse performance than other methods).

Thank you for highlighting these points, we believe that they have been addressed in full and feel that the changes really improved the quality of the manuscript. 

Reviewer #3: 

Comment 1: The proposal of the manuscript is very interesting. 

Answer 1: We thank the reviewer for this positive evaluation.

Comment 2: However, more detailed information about the analytical methods applied is important. For example, this data could be added as Supplementary Material: - ions used for quantification and qualification (and retention time) of analytes analyzed by GC-MS and LC-MS.

Answer 2: This information was added as Supplementary Tables. In addition, we have added a reference to another paper describing the analytical methods used (https://link.springer.com/article/10.1007/s00216-016-9933-x, see line 170). 

Comment 3: - more details about the materials used in the study such as solvents, reagents, chemicals, and standards, including manufacturers, purity, solutions preparation...

Answer 3: Thank you for your comment, we added these details to the manuscript and to the Supplementary Tables.

Comment 4: - Page 6 (line 114): What is the sample amount used for extraction?

Answer 4: The amounts of pooled BAL samples and urine used for extraction were added (see lines 108-113, 131 and 155).

Comment 5: - Page 7 (line 137): "using a modified CDC protocol 6705.02 (19)": add a brief description of the protocol.

Answer 5: We have added a more detailed description (see lines 155-162).

Comment 6: - Page 7 (line 143): " Analyte detection was achieved by means of tandem mass spectrometer AB Sciex QTrap 5500 operating in negative electrospray ionization mode...": add information about the colision energy used to obtain the quantification and qualification ions.

Answer 6: We have improved the information on the method and added a reference to a paper containing a detailed procedure (https://link.springer.com/article/10.1007/s00216-016-9933-x, see line 170). 

Comment 7: - Data about the performance characteristics of the analytical methods is very important such as limit of detection, limit of quantification, linearity, recovery and precision.

Answer 7: The part on the performance of the method in the Chapter 2.3 was extended, LOD/LOQ were added as Supplementary Tables.

---

## [Editor Report · Decision Letter 1]

12 Jul 2024

PONE-D-24-10533R1Polycyclic aromatic hydrocarbons and their metabolites in bronchoalveolar lavage and urine samples from patients with inhalation injury throughout their hospitalization: a prospective pilot studyPLOS ONE

Dear Dr. Borilova Linhartova,

Thank you for submitting your manuscript to PLOS ONE. After careful consideration, we feel that it has merit but does not fully meet PLOS ONE’s publication criteria as it currently stands. Therefore, we invite you to submit a revised version of the manuscript that addresses the points raised during the review process.

After my personal, careful evaluation of your revised version, and of the response provided to the reviewers, I think that your manuscript would be suitable for publication provided that the following minor change is made:

- remove the reference 19 inserted at the request of one of the reviewers, which is very marginally related to the manuscript and does not introduce any substantial improvement to the manuscript.

We look forward to receiving your revised manuscript.

Kind regards,

Giovanni Signore

Academic Editor

PLOS ONE
---

## [Author Response · Author response to Decision Letter 1]

16 Jul 2024

Response to Reviewers (Editor)

Editor´s Comment 1: After my personal, careful evaluation of your revised version, and of the response provided to the reviewers, I think that your manuscript would be suitable for publication provided that the following minor change is made:

- remove the reference 19 inserted at the request of one of the reviewers, which is very marginally related to the manuscript and does not introduce any substantial improvement to the manuscript.

Journal Requirement 1: Please review your reference list to ensure that it is complete and correct. If you have cited papers that have been retracted, please include the rationale for doing so in the manuscript text, or remove these references and replace them with relevant current references. Any changes to the reference list should be mentioned in the rebuttal letter that accompanies your revised manuscript. If you need to cite a retracted article, indicate the article’s retracted status in the References list and also include a citation and full reference for the retraction notice.

Answer 1: We removed the reference required by reviewer from the manuscript. We really appreciate your insight and suggestion. All references and quality of figures were checked.

Rabieimesbah A, Sobhanardakani S, Cheraghi M, Lorestani B. Concentrations, Source Identification and Potential Ecological and Human Health Risks Assessment of Polycyclic Aromatic Hydrocarbons (PAHs) in Agricultural Soils of Hamedan County, West of Iran. Soil Sediment Contam Int J. 2024;33: 482–506. 

Journal Requirements 2: While revising your submission, please upload your figure files to the Preflight Analysis and Conversion Engine (PACE) digital diagnostic tool, https://pacev2.apexcovantage.com/. PACE helps ensure that figures meet PLOS requirements. To use PACE, you must first register as a user. Registration is free. Then, login and navigate to the UPLOAD tab, where you will find detailed instructions on how to use the tool. If you encounter any issues or have any questions when using PACE, please email PLOS at figures@plos.org. Please note that Supporting Information files do not need this step.

Answer 2: Thank you for this recommendation. All figures from the main text have been uploaded to PACE tool and, thus, they should be meeting PLOS requirements.

---

## [Editor Report · Decision Letter 2]

18 Jul 2024

Polycyclic aromatic hydrocarbons and their metabolites in bronchoalveolar lavage and urine samples from patients with inhalation injury throughout their hospitalization: a prospective pilot study

PONE-D-24-10533R2

Dear Dr. Borilova Linhartova,

We’re pleased to inform you that your manuscript has been judged scientifically suitable for publication and will be formally accepted for publication once it meets all outstanding technical requirements.

Kind regards,

Giovanni Signore

Academic Editor

PLOS ONE
---

## [Editor Report · Acceptance letter]

24 Jul 2024

PONE-D-24-10533R2 

PLOS ONE

Dear Dr. Borilova Linhartova, 

I'm pleased to inform you that your manuscript has been deemed suitable for publication in PLOS ONE. Congratulations! Your manuscript is now being handed over to our production team.

Kind regards, 

on behalf of

Dr. Giovanni Signore 

Academic Editor

PLOS ONE